# 'Microincisional trabeculectomy for glaucoma"

**Aparna Rao**  *, **Sardar Khan, Sujoy Mukherjee**

Glaucoma Service, LV Prasad Eye Institute, Patia, Bhubaneswar, India

* aparna@lvpei.org, vinodini10375@yahoo.com

## Abstract

### Purpose

To evaluate the short-term clinical outcomes of microincisional trabeculectomy (MIT), a new technique of ab-interno trabeculectomy.

### Methods

Consecutive patients with open-angle glaucoma identified from the hospital database that underwent MIT with or without cataract surgery between September 2021 to June 2022 at a tertiary eye centre in East India, were screened. Those with a follow-up of < 6 months or with incomplete data were excluded. MIT was done ab-interno using microscissors and microforceps in 2–4 clock hours of the nasal angle via a temporal incision. The intraocular pressure (IOP) reduction at 6 months, and reduction in the number of medications after surgery were analysed. Surgical success (IOP>6 and <22 mm Hg), complications, angle features on anterior segment optical coherence tomography (ASOCT), and the need for additional surgeries were analysed.

### Results

We included thirty-two eyes of 32 patients with open-angle glaucoma (including n = 9 eyes that underwent concurrent cataract surgery) with a preoperative mean IOP of 22 ±11.1 mm Hg and visual field index of 47±37.9%. All eyes achieved >30% IOP reduction, with a final IOP of 14±6.9 mm Hg at 6 months. Surgical success in 31 of 32 eyes with complete success seen in 28 eyes with none of the eyes requiring >1 medication for IOP control. Hyphema was seen in 4 eyes, while transient IOP spikes at 1 day-1 month were seen in 5 eyes, none of which required any additional interventions. One eye with persistent raised IOP at 1 month required incisional trabeculectomy for uncontrolled IOP with 2 medications.

### Conclusion

MIT, a new technique of ab-interno trabeculectomy, is effective in terms of IOP control and reduction in the number of medications while having fewer complications. Long-term studies comparing the efficacy of MIT with incisional trabeculectomy, or other procedures are warranted in the future.

**Data Availability Statement:** All relevant data are within the paper and its Supporting Information files. A supporting file with minimal data has been provided as suggested.

**Funding:** The author(s) received no specific funding for this work.

**Competing interests:** The authors have declared that no competing interests exist.

## Introduction

Newer surgical procedures like minimally invasive glaucoma surgery (MIGS) over the past decade are now increasingly being adopted for various clinical situations in glaucoma [1–6]. These procedures offer the possibility of sparing the conjunctiva for future filtering surgeries while providing good intraocular pressure (IOP) control and reducing the need for glaucoma medications [2, 4, 5]. Apart from newer ab-interno glaucoma stents, a gamut of MIGS procedures includes goniotomy, suture or catheter gonioscopy assisted transluminal trabeculotomy (GATT), bent angled needle goniotomy (BANG) and trabectome. Though all MIGS procedures are not maximally effective in all cases or situations, these have now revolutionised glaucoma surgery, especially after the COVID-19 pandemic, and are slowly replacing trabeculectomy, at least in specific cases [1, 4–9].

Trabeculectomy remains the surgery of choice in most clinical situations, in glaucoma practice apart from being a reliable and dependable procedure [6]. However, the complications associated with this procedure including hypotony, choroidal detachment and bleb-related infections, have forced surgeons to seek safer alternatives to trabeculectomy. MIGS procedures are now being adopted increasingly by many surgeons across the world [2–5, 7–9]. Surgical approaches in glaucoma have therefore shifted focus to the Schlemm's canal and the collector channels in addition to the trabecular meshwork (TM), which are specifically targeted by MIGS [7, 9]. Though the long-term results in terms of IOP control are varied for different procedures, targeting the TM and the SC helps achieve >20% IOP reduction, with fewer complications rates than trabeculectomy. Yet, trabeculectomy still remains as the most important rescue for most surgeries when MIGS fails or are contraindicated in special conditions [6]. There is a need for devising a clean, easy and effective means of bleb-less trabeculectomy that is not associated with bleb-related or hypotony related complications of ab-externo incisional trabeculectomy. It should also have a good safety profile and fair to good outcomes under routine clinical situations. We herein describe a technique of ab-interno trabeculectomy that is not only effective but also a safe, easy, and effective procedure for most clinical situations as a stand-alone procedure or in combination with other surgeries.

## Materials & methods

All patients with open-angle glaucoma with uncontrolled intraocular pressure or intolerant to medications attending glaucoma services at a tertiary eye care setting between September 2021-June 2022, were identified using the hospital electronic medical record database. The study followed the tenets of the declaration of Helsinki and was approved by the Institutional review board of LV Prasad Eye Institute, Bhubaneswar (IEC-16-IM-3). Patients were included if they had undergone microincisional trabeculectomy alone with or without cataract surgery in the stated period. Patients with at least 6 months of follow-up after surgery were included into the study after obtaining an informed written consent, as per institutional protocol. Patients with <6 months follow-up, closed angles, secondary glaucoma including neovascular glaucoma, those that underwent complete circumferential gonioscopic circumferential transluminal trabeculotomy or ab-externo trabeculectomy earlier, patients undergoing additional retinal surgeries/injections simultaneously with glaucoma surgery for ocular associations, and nonconsenting patients, were excluded.

### The microincisional trabeculectomy-technique

All surgeries were performed in the nasal quadrant via a temporal incision (surgeon seated temporally) by the same surgeon (APR) under local anaesthesia using a Swan-Jacob lens for angle visualization. For patients undergoing combined glaucoma and cataract surgery, the

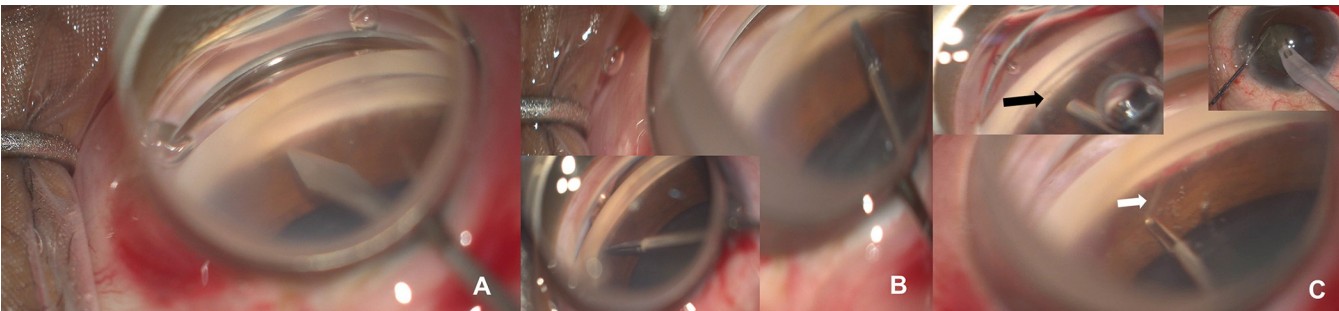

**Fig 1.** A-C show the procedure of MIT that involves a goniotomy in the nasal angle (through temporal incision) using an MVR blade (A) followed by (B) vertical incisions on the trabecular meshwork over 2–4 clock-hours using 25- gauge straight vitreoretinal microscissors-note the position of the scissors (note vertical positions of the scissors in the inset). C show the stripping of the cut trabecular meshwork (white arrow) using an end-gripping microforceps with the left inset showing the opened trabecular shelf (black arrow) in the specific clock hours which is followed by a phacoemulsification temporally (top right inset).

MIT was done before phacoemulsification. The procedure required a goniotomy lens to visualize the angle directly under the operating microscope. After goniotomy with an MVR microsurgical blade, the Schlemm's canal was identified followed by two vertical (oriented perpendicular to the TM-see Fig 1) cuts given using the straight vitreoretinal 25-gauge microscissors over 2–4 clock hours of the angle, Fig 1. Now, the delineated TM strip over the 2–4 clock hours of the angle between the two vertical cuts, was dissected from the shelf using a 25-gauge end-gripping vitreoretinal forceps. This was now stripped off till the other cut end in one stroke, Fig 1. Care was taken during this step to align the scissors vertically over the TM to avoid trauma to the iris below. Now, the delineated TM strip over the 2–4 clock hours of the angle between the two vertical cuts, was dissected from the shelf using a 25-gauge end-gripping vitreoretinal forceps. This was now stripped off till the other cut end in one stroke by a gentle pulling force with the forceps, Fig 1. Any bleeding during this procedure was handled by tamponade with viscoelastic injection into the chamber. The TM tissue that was stripped off was now removed and cataract surgery, if required, was completed through a temporal corneal incision. Intracameral pilocarpine was injected at the end of the surgery after a thorough wash of the viscoelastic, and the chamber was filled with air at the end of the procedure.

The postoperative regimen for these patients included topical steroids tapered slowly over 1 month. The patients were followed up at 1 week, 4 weeks, 6 months until the last follow-up. Any intra- or postoperative complications were noted. IOP spikes, defined as >21mm Hg at any period after surgery, were treated with short course of acetazolamide inhibitors for one week followed by topical medications for persistent spikes.

Anterior segment Optical Coherence Tomography (Swept source OCT, Triton TM DRI, Topcon, USA) was performed pre-operatively, 1 month, and at the onset of postoperative IOP spike in cases where this occurred. All 4 quadrants were imaged in a standard manner and under standard lighting conditions in non-dilated state [10]. The degrees of open TM shelf, presence of peripheral anterior synechiae (PAS), supraciliary effusions/cleft or clots, or additional angle features, were noted and correlation was done with gonioscopic findings. Transient IOP spikes were defined as rise in IOP >3 mm Hg from 1day-4 weeks of the pre-surgery IOP without the addition of medications.

Success was defined as IOP >6 mm Hg and <22 mm Hg (complete if this was achieved without the need for medicines or qualified with set IOP achieved after addition of medications) without the need for additional surgeries in the postoperative period. Hypotony was defined as an IOP <6 mm Hg with structural changes in the retina/optic nerve with loss of visual acuity of ≥2 Snellen lines. Failure was defined as need for additional surgeries, loss of

vision due to any reason after surgery, or uncontrolled IOP>22 mm Hg despite anti-glaucoma medications at 1month.

## Statistics

Demographic and clinical data are represented in descriptive manner with normative data being presented as mean and standard deviation (SD), while non-normally distributed data are presented as median and range. Continuous data were compared using the paired Student's t test or the Wilcoxon test. All analysis was done using Stata Corp (USA, version 14) with statistical significance was set at p<0.005. Analysis was repeated with IOP criteria being set to <18 mm Hg and results compared to those obtained with standard criteria for surgical success. Results were also compared between eyes that underwent MIT alone or combined with cataract surgery.

## Results

Of a total of 36 eyes of 36 patients who underwent MIT during the study period, 4 were excluded owing to additional procedures like vitreo-retinal procedures being performed in the same sitting. A total of 32 eyes of 32 patients were included (n = 9 undergoing concurrent cataract surgery) with a mean age of 60±13.5 years and a mean follow-up of 8±1.9 months. The baseline clinical profile of the patients is detailed in **Table 1**.

The preoperative visual acuity was worse in eyes undergoing cataract surgery with MIT, though the visual field indices were not statistically different, **S1 Table**. The median age, IOP at the time of surgery and the number of anti-glaucoma medications were not significantly different between the two groups, S1 Table.

Table 2 and Fig 2 shows the IOP profiles of the patients. All eyes achieved a significant reduction in the number of medications at 1 month that persisted at 6 months (p<0.001 for all periods compared to preoperative IOP), with 1 eye requiring incisional trabeculectomy at 1 month for uncontrolled IOP. Transient IOP spikes at <2 weeks after surgery were seen in 5 eyes; four eyes developed IOP spikes immediately (1d-1week) after surgery due to retained viscoelastic, that responded well to a short course of acetazolamide inhibitors for 2–3 days.

**Table 1. Baseline demographic and clinical profile of patients that underwent microincisional trabeculectomy (MIT)- see text for full description.**

| Variables | Mean ± standard deviation N = 32 |
|---|---|
| Age (years) | 60±13.5 |
| Male:Female | 7:25 |
| Preop VA (decimals) | 0.2±0.2 |
| IOP at the time of surgery (mm Hg) | 22±11.1 |
| Number of medications before surgery | 3±1.2 |
| Diagnosis | Primary open-angle glaucoma-21 Pseudoexfoliation glaucoma -8 Juvenile open-angle glaucoma-2 Angle recession-1 |
| Mean deviation (dB) | -18±10.9 |
| Visual field index (%) | 47±37.9 |
| Cataract surgery performed concurrently (n) | 9 |

VA- visual acuity, IOP-intraocular pressure

**Table 2. Postoperative clinical outcomes after microincisional trabeculectomy (MIT)-see text for full description.**

| Variables | Mean ± standard deviation N = 32 |
|---|---|
| IOP week 1 (mm Hg)* | 13±4.5 |
| IOP 1month (mm Hg)* | 15±5.03 |
| IOP 6 months (mm Hg)* | 14±6.9 |
| Percentage reduction of IOP (%) | 32–58 |
| Number of eyes requiring medication for IOP control 2 medications 1 medication | 4 1 3 |
| Hyphema | 4 |
| Number of eyes having transient IOP spikes | 5 |
| Repeat procedures/surgery | 1 |
| Hyperreflective membrane on ASOCT | 5 |
| Loss of vision | 0 |

IOP-intraocular pressure; ASOCT- Anterior segment Optical Coherence Tomography, *-all IOPs statistically different from preoperative IOP.

Hyphema was seen in 4 eyes that resolved spontaneously without the need for wash or additional procedures.

Four eyes required anti-glaucoma medications for IOP of >22 mm Hg with one of the eyes requiring >2 medicines, Table 2. One eye with angle recession glaucoma had high IOP at 1 month after surgery, that was uncontrolled despite addition of 2 medications, mandating an incisional trabeculectomy. The final IOP at 6 months or rate of complications was not significantly different between eyes that underwent MIT with or without cataract surgery, **S2 Table**.

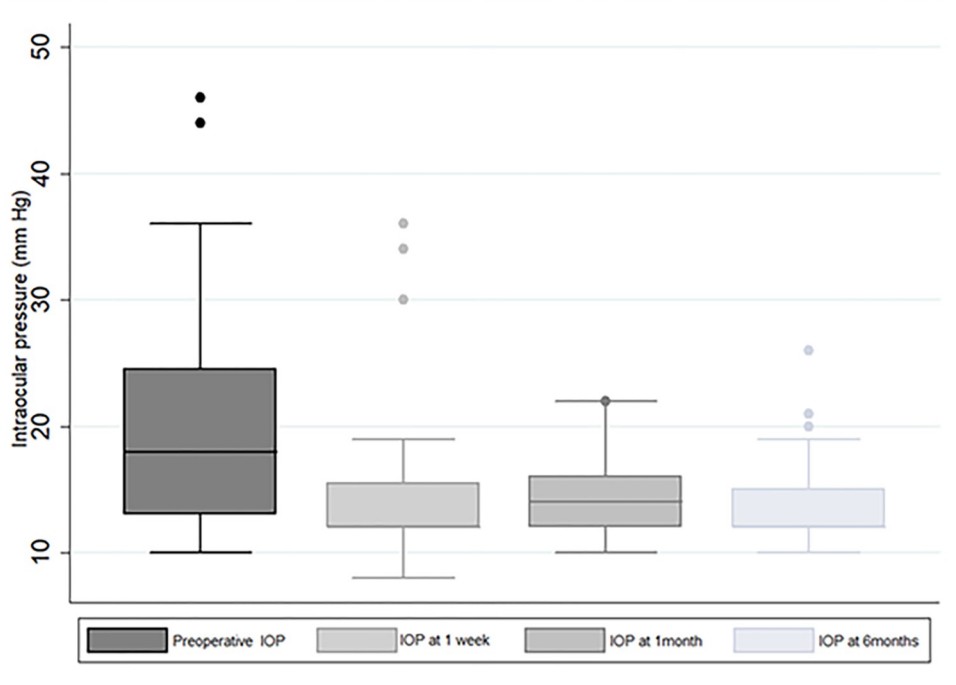

**Fig 2. Box-whisker plots showing IOP profiles after microincisional trabeculectomy (MIT)-see text for full description.**

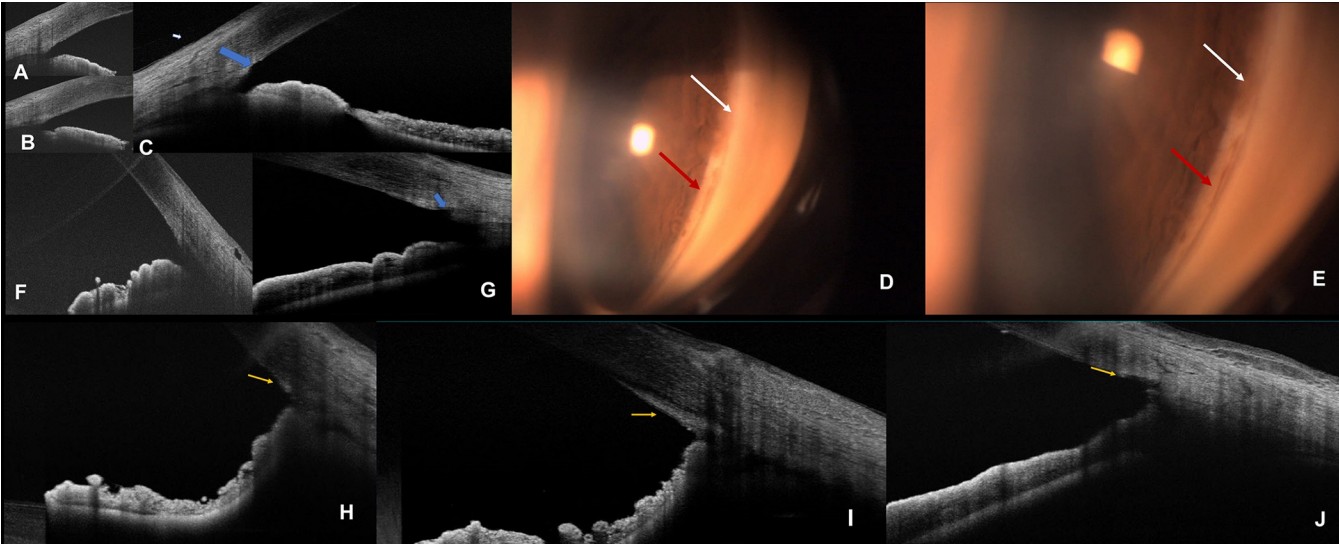

**Fig 3.** A-C shows the angle of a patient with open-angle glaucoma that underwent MIT preoperatively (A, B) and postoperatively (C) showing the saucerization at the trabecular meshwork region (blue arrows). D and E depict the gonioscopic finding in the patient showing open bare Schlemm's canal in specific clock-hours (red arrow) after MIT while the intact untouched TM is visible adjacently (white arrow). F and G shows similar findings in another patient with open angles (D) showing the saucerised TM (E, blue arrows). H, I and J show the ASOCT feature of hyperreflective membrane over the trabecular meshwork region (yellow arrows).

On ASOCT, the region where TM was removed appeared as a saucerization of the TM region in the specific clock-hours, with no other associated findings of a separate trabecular leaflet, Fig 3. A hyperreflective membrane (HM) over 0.5–1 clock hour was seen (within the same regions of removal of the TM) on ASOCT in 5 eyes, suggestive of partial closure of the open SC by a fibrotic membrane, with none of these eyes requiring additional medications or raised IOP. None of the eyes had an IOP<6 mm Hg, or experienced loss of vision. One eye required additional/repeat procedures in the form of incisional trabeculectomy at 1month despite obvious saucerization seen on ASOCT.

Surgical success was therefore seen in 31 of 32 eyes (complete success in 28 eyes, qualified success in 3 eyes). Failure was seen in therefore in one eye with angle recession, that required an incisional trabeculectomy.

## Discussion

Microincisional trabeculectomy, a form of ab-interno limited trabeculectomy procedure, seemed to achieve 30–58% reduction of IOP with absolute reduction in the number of medications when used as an isolated procedure, or when combined with cataract surgery. It also had fewer transient complications that did not need additional procedures or surgery. Moreover, it does not require sophisticated instruments/machines or stents, and minimizes collateral damage or hypotony related issues, common for incisional routine ab-externo trabeculectomy. Though a long-term outcome >6 months is desirable for any glaucoma surgery, we believe the results are encouraging and comparable to short-term IOP results of other MIGS procedures with fewer complications.

MIT removes the TM in specific portions of the angle using clean cuts with microscissors, while not causing ablative, thermal injury, or forceful ripping, as seen in GATT or goniotomy using the Kahook-dual blade (KDB) blade. The clean cuts ensure minimal bleeding, minimizes trauma to adjacent tissues like iris, improves outflow in focused portion of the angle (avoiding

damage with circumferential damage to the whole angle with functional healthy TM in some sectors) while ensuring a significant opening of the trabecular shelf with the least trauma. This may be the reason we did not see high rates of hyphema or raised IOP spikes as seen with other similar MIGS procedures like GATT, while achieving >30% IOP reduction. While identifying the specific portion of the angle (or TM) to be targeted in any glaucoma surgery is indeed a scientifically plausible futuristic goal, MIT still allows repeatability, is easy & efficacious in circumventing the resistance in the conventional outflow pathway, obviates need for expensive instruments, and is minimally invasive in a true sense.

Recently, there has been an increasing trend of adopting MIGS into glaucoma practice. Breakthrough insights into understanding of the outflow pathway [1–5, 8, 9, 11–15] are the reason behind the rising popularity of MIGS. This shows that there is targeted focus on enhancing aqueous outflow at the site of maximum concentration of collector channels, by bypassing or ablating the site of outflow resistance, the trabecular meshwork. These procedures are gaining momentum resulting in increasing adoption of MIGS as primary surgery in glaucoma management. The increasing wave of MIGS procedures in glaucoma management is understandable. They allow for a bleb-less filtering surgery that obviates the catastrophic complications seen with filtering incisional trabeculectomy [4, 5]. Though trabeculectomy still remains the most recalled and well-known rescue in most clinical situations, MIGS now is slowly superseding trabeculectomy as the primary procedure in most forms of glaucoma across the world [3–6]. We believe MIT will also have its place as a safe, easy, and efficacious surgery among other MIGS procedures, with potential to be adopted as a primary surgical option for various clinical situations in glaucoma, given the safety profile of the procedure.

The success rates in terms of IOP control with MIGS is reported to be comparable to filtering procedures, with most offering IOP and safety profiles compared to conventional filtering surgery [1, 3, 4, 5, 10–16]. Yet, this evidence comes mostly from non-comparative studies. The MIGS procedures so far include ways to remove the TM by thermal/laser ablation or simply bypass the TM by incising the Schlemm's canal (goniotomy), an affect that was shows many years earlier in enucleated eyes [8, 9]. The trabectome has shown good promise when combined with cataract surgery and is now slowly being superseded by GATT and 360 degree suture or microcatheter trabeculotomy [4, 5, 6, 14–16]. GATT has gained popularity over the last 5 years because of its ease, cost effectiveness and efficacy in terms of IOP control, though it has a steep learning curve. Further, hyphema and IOP spikes are very frequent complications seen after GATT [14–16]. The Kahook dual blade (KDB) goniotomy procedure has similar efficacy similar to other MIGS or ab-externo goniotomy in open-angle or childhood glaucoma [3–5, 17, 18]. These aforementioned procedures either dilate the Schlemm's canal forcefully using specially designed devices such as an angled KDB blade, suture, or microcatheter and/or involve removing the TM by thermal ablation or by forceful ripping/pulling of the suture/microcatheter [3, 15–18]. In contrast, MIT opens up the SC in a non-traumatic manner with minimal bleeding and reduces the chances of sudden changes in pressure dynamics within the SC, that can cause collapse of the collector channels. It also restricts the opening of the trabecular shelf in specific quadrants with an ab-interno approach. This makes it repeatable, spares the conjunctiva, and limits the trauma to adjacent functional areas of the TM as seen with GATT or other circumferential procedures. Restricting TM removal to small quadrants with adequate IOP control in a non-traumatic manner akin to conventional trabeculectomy, while obviating the need for special devices/ blades/stents, suggests that this procedure that can be adopted by most surgeons irrespective of the availability for specific devices/expensive implants, or instruments.

Angle dysgenesis on gonioscopy is described in exfoliation glaucoma and juvenile open-angle glaucoma patients [19, 20]. A hyperreflective membrane on ASOCT suggests an aberrant

TM tissue and outflow pathway, that is presumed to the mechanism of glaucoma in these patients [19]. The HM seen in this study may reflect "aberrant TM" and is indicative of resultant activation of fibroblastic response in the region of MIT. Yet, none of these eyes required any medications for IOP control, implying that a clean cut to the TM with microscissors is less traumatic and incites a less aggressive fibroblastic response than tearing/ripping the TM tissue as seen in GATT or goniotomy. Failure in one eye requiring incisional trabeculectomy also showed an open SC possibly suggesting that the cause of failure was damaged collector channels in that eye rather than failure of the procedure itself.

There are few limitations to this study apart from those that are inherent with a retrospective study design. We have not compared this procedure with other MIGS procedures like GATT or with incisional trabeculectomy in this pilot study. We did not include angle closure eyes or eyes with early glaucoma possibly because of bias introduced by need for glaucoma surgery required mostly in later glaucoma stages. A randomized trial between MIT, GATT, and external trabeculectomy is warranted for validating our results in future studies. The success of any glaucoma surgery requires long-term term results. The long-term outcomes and visual function changes with cost effectiveness will lend credence to the efficacy of this procedure in isolation or when combined with cataract surgery in the future.

## Conclusions

This study finds MIT, an easy method of ab-interno trabeculectomy, to be a potential addendum to the list of MIGS procedures. It is minimally invasive and maximally effective in terms of IOP control over 6 months, with an absolute reduction in the number of glaucoma medications. Long-term studies comparing this procedure versus conventional trabeculectomy or routinely practiced procedures like GATT, is warranted in future.

## Supporting information

**S1 Table. Comparison of clinical profile of patients undergoing either MIT alone or combined with cataract surgery.**
(DOCX)

**S2 Table. Comparison of IOP outcomes and complications rates between eyes that underwent MIT alone or combined with cataract surgery.**
(DOCX)

**S1 Data.**
(XLSX)

## Acknowledgments

Hyderabad Eye Research Foundation.

## Author Contributions

**Conceptualization:** Aparna Rao.

**Data curation:** Aparna Rao, Sujoy Mukherjee.

**Formal analysis:** Aparna Rao, Sardar Khan.

**Investigation:** Aparna Rao, Sardar Khan, Sujoy Mukherjee.

**Methodology:** Aparna Rao, Sardar Khan, Sujoy Mukherjee.

**Project administration:** Aparna Rao, Sardar Khan, Sujoy Mukherjee.

**Software:** Aparna Rao.

**Supervision:** Aparna Rao, Sujoy Mukherjee.

**Validation:** Aparna Rao.

**Writing – original draft:** Aparna Rao, Sardar Khan.

**Writing – review & editing:** Aparna Rao, Sardar Khan, Sujoy Mukherjee.

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
