## [Decision Letter · Decision Letter 0]

7 Mar 2023

PONE-D-23-01259‘Microincisional trabeculectomy for glaucoma”PLOS ONE

Dear Dr. Rao,

Thank you for submitting your manuscript to PLOS ONE. After careful consideration, we feel that it has merit but does not fully meet PLOS ONE’s publication criteria as it currently stands. Therefore, we invite you to submit a revised version of the manuscript that addresses the points raised during the review process.

A reviewer have suggested a few minor changes that can be easily incorporated in a revised manuscript. 

We look forward to receiving your revised manuscript.

Kind regards,

Sanjoy Bhattacharya

Academic Editor

PLOS ONE

Journal Requirements:

“Hyderabad Eye Research Foundation”

Reviewers' comments:

Reviewer's Responses to Questions

**Comments to the Author**

1. Is the manuscript technically sound, and do the data support the conclusions?

Reviewer #1: Yes

Reviewer #2: Yes

2. Has the statistical analysis been performed appropriately and rigorously? 

Reviewer #1: Yes

Reviewer #2: N/A

3. Have the authors made all data underlying the findings in their manuscript fully available?

Reviewer #1: Yes

Reviewer #2: Yes

4. Is the manuscript presented in an intelligible fashion and written in standard English?

Reviewer #1: Yes

Reviewer #2: Yes

5. Review Comments to the Author

Reviewer #1: This project targets an interesting question and provides new insight into the field. It would benefit from more measured variables such as follow-up on post-op visual fields and visual acuities if possible. The provided data on rates of complications does seem to suggest that this study merits larger follow-up studies to confirm your findings. You should add the specific software and packages you used to perform your statistics.

Reviewer #2: 1) Generally, I believe that the manuscript was well-written, clear, organized, and concise. I do not have much experience in statistical analysis so I am unable to comment on this matter, but from what I see, the tables, figures, and descriptions presented in the manuscript were clear and easily understandable. However, I think it may be more clear if the figures were located in the results section for easy reference (unless they are being considered as supplemental tables). If not placed in the results section, it would be easier to understand if legends were placed directly below the figures and tables.

2) There were a few grammatical/punctuational issues present, including:

Abstract:

"1day-1month" -> insert spaces to "1 day-1 month"

Materials & Methods:

"LV prasad eye institute" -> capitalize to "LV Prasad Eye Institute"

"followed up at 1 week, 4 weeks, 6months" -> insert space to "6 months"

Discussion:

"increasing wave of MIGS procedures in glaucoma management is understandable They" -> insert period to "is understandable. They"

"The Kahook dual blade (KDB) goniotomy procedure is has similar efficacy to other" -> reword to "goniotomy procedure has a similar efficacy to other"

Table 2:

"2mediciations" -> spelling error + insert space to "2 medications"

3) In both figures 1 and 3, the letters corresponding to specific to each image (A, B, C, etc.) could be larger and more clear to the reader. The letters seem to be a bit small and difficult to locate.

Overall, this is an exciting finding and authors did a great job with presentation of results and explaining details concisely. Some improvements to the value of this study include another follow up after 6 months and increasing the sample size.

6. PLOS authors have the option to publish the peer review history of their article (what does this mean?). If published, this will include your full peer review and any attached files.

Reviewer #1: No

Reviewer #2: **Yes: **Jessica Lee

---

## [Author Response · Author response to Decision Letter 0]

8 Mar 2023

Reviewer #1: This project targets an interesting question and provides new insight into the field. It would benefit from more measured variables such as follow-up on post-op visual fields and visual acuities if possible. The provided data on rates of complications does seem to suggest that this study merits larger follow-up studies to confirm your findings. You should add the specific software and packages you used to perform your statistics.

Answer: We thank the reviewer for the encouragement of our work. The reviewer is very correct and bang on the actual benefits of any surgical procedure for glaucoma being the preservation of long-term visual function. This is exactly the reason why we are now following the patients with visual fields in a separate longitudinal study comparing MIT with trabeculectomy and GATT. The visual fields are usually done on an annual basis in India which is why we did not feel adding a 6months visual field data, would be informative with regards to the long-term visual field progression rates with each surgical procedure. We definitely will update the reviewers and readers on the long-term results in our future papers or even on a personal request. We apologize for not mentioning the software used for analysis in this study. We have now added this information, as suggested. 

Reviewer #2: 1) Generally, I believe that the manuscript was well-written, clear, organized, and concise. I do not have much experience in statistical analysis so I am unable to comment on this matter, but from what I see, the tables, figures, and descriptions presented in the manuscript were clear and easily understandable. However, I think it may be more clear if the figures were located in the results section for easy reference (unless they are being considered as supplemental tables). If not placed in the results section, it would be easier to understand if legends were placed directly below the figures and tables.

Answers: We are very happy with the reviewer’s comments and pleased to share all the results of this new technique and any results of our future studies on this procedure on even a personal request. We would like to clarify that have made the figures separately available and not in the word document since adding them to the document decreases their resolution which is very crucial for understanding this procedure correctly. Also, the format of the manuscript is as per Plos One requirements which require files to be submitted separately with legends as per Plos One author instructions. 

2) There were a few grammatical/punctuational issues present, including:

Abstract:

"1day-1month" -> insert spaces to "1 day-1 month"

Materials & Methods:

"LV prasad eye institute" -> capitalize to "LV Prasad Eye Institute"

"followed up at 1 week, 4 weeks, 6months" -> insert space to "6 months"

Discussion:

"increasing wave of MIGS procedures in glaucoma management is understandable They" -> insert period to "is understandable. They"

"The Kahook dual blade (KDB) goniotomy procedure is has similar efficacy to other" -> reword to "goniotomy procedure has a similar efficacy to other"

Table 2:

"2mediciations" -> spelling error + insert space to "2 medications"

3) In both figures 1 and 3, the letters corresponding to specific to each image (A, B, C, etc.) could be larger and more clear to the reader. The letters seem to be a bit small and difficult to locate.

Overall, this is an exciting finding and authors did a great job with presentation of results and explaining details concisely. Some improvements to the value of this 

Answer: We thank the reviewer for the suggestions and all typos have been corrected in the revised manuscript. We have also made the letters in the figures bigger for easy visibility, as suggested.

---

## [Decision Letter · Decision Letter 1]

25 Apr 2023

PONE-D-23-01259R1‘Microincisional trabeculectomy for glaucoma”PLOS ONE

Dear Dr. Rao,

Thank you for submitting your manuscript to PLOS ONE. After careful consideration, we feel that it has merit but does not fully meet PLOS ONE’s publication criteria as it currently stands. Therefore, we invite you to submit a revised version of the manuscript that addresses the points raised during the review process.

A learned reviewer have recommended description of statistical software and some related details, which must be included in a revised manuscript. 

We look forward to receiving your revised manuscript.

Kind regards,

Sanjoy Bhattacharya

Academic Editor

PLOS ONE

Journal Requirements:

Reviewers' comments:

Reviewer's Responses to Questions

**Comments to the Author**

1. If the authors have adequately addressed your comments raised in a previous round of review and you feel that this manuscript is now acceptable for publication, you may indicate that here to bypass the “Comments to the Author” section, enter your conflict of interest statement in the “Confidential to Editor” section, and submit your "Accept" recommendation.

Reviewer #1: All comments have been addressed

Reviewer #2: All comments have been addressed

2. Is the manuscript technically sound, and do the data support the conclusions?

Reviewer #1: Yes

Reviewer #2: Yes

3. Has the statistical analysis been performed appropriately and rigorously? 

Reviewer #1: Yes

Reviewer #2: N/A

4. Have the authors made all data underlying the findings in their manuscript fully available?

Reviewer #1: Yes

Reviewer #2: Yes

5. Is the manuscript presented in an intelligible fashion and written in standard English?

Reviewer #1: Yes

Reviewer #2: Yes

6. Review Comments to the Author

Reviewer #1: Thank you for addressing our previous comments. The paper has made significant progress, but still requires some minor editing.

Methods

"student-t test" should be Student's t-test.

Results

There is a line where you need a space between '6' and 'months'. "Hyphema was seen in 4 eyes that resolved spontaneous." Spontaneous should be edited to spontaneously. You switch between writing numbers numerically and spelling them out, please pick a format for consistency.

Discussion

There are some sentences that could use minor editing. "The Kahook dual blade (KDB)

goniotomy procedure has similar efficacy similar to other MIGS or ab-externo

goniotomy in open-angle glaucoma, an2mediciations d primary congenital

glaucoma [3-5,17,18]." Please revise here

Reviewer #2: Authors completed all revisions and looks good. Grammar changes were made, and figures are more clear with clear labels.

7. PLOS authors have the option to publish the peer review history of their article (what does this mean?). If published, this will include your full peer review and any attached files.

Reviewer #1: No

Reviewer #2: No

---

## [Author Response · Author response to Decision Letter 1]

29 Apr 2023

¬¬To,

The Editor,

Dear Sir/Madam,

We hereby re-submit our revised manuscript “Microincisional trabeculectomy for glaucoma “along with the point-point clarification to the reviewer’s comments. We believe the suggestions and corrections were very apt and would welcome further suggestions. We have also professionally edited this manuscript. 

All the authors have contributed equally towards the preparation of the manuscript and have no financial or proprietary interest in the products used in the study. We also declare that this article has not been published previously or is under review with any other journal.

a) All acknowledgments and financial disclosures/funding information is included in the manuscript. We would like to clarify that this acknowledgment is for the research foundation that supports all healthcare-related studies at the institute. This does not entail funding from the foundation. This study did not receive any funding from any agency or organization. 

b) All data have been given in the manuscript with additional patient-identifying information that may be shared after consent upon request. We have also made available a minimal dataset along with all relevant data that has already been shared in the supplemental data. 

Thanking you

Reviewer #1: Thank you for addressing our previous comments. The paper has made significant progress, but still requires some minor editing.

Methods

"student-t test" should be Student's t-test.

Results

There is a line where you need a space between '6' and 'months'. "Hyphema was seen in 4 eyes that resolved spontaneous." Spontaneous should be edited to spontaneously. You switch between writing numbers numerically and spelling them out, please pick a format for consistency.

Discussion

There are some sentences that could use minor editing. "The Kahook dual blade (KDB) goniotomy procedure has similar efficacy similar to other MIGS or ab-externo

goniotomy in open-angle glaucoma, an2mediciations d primary congenital

glaucoma [3-5,17,18]." Please revise here

Answers: We thank for the exhaustive review of important typos and we really feel sorry for errors like Student’s t test which we feel was not appropriate. We have corrected this and other typo errors. We would be ready to correct any other typos if any found still in the revised manuscript. 

Reviewer #2: Authors completed all revisions and looks good. Grammar changes were made, and figures are clearer with clear labels.

Answer: we thank the reviewer for the encouraging words and believe that the follow up of this paper may change the way clinician scientists look at the tissue in glaucoma.

Review Comments to the Author

Reviewer #1: This project targets an interesting question and provides new insight into the field. It would benefit from more measured variables such as follow-up on post-op visual fields and visual acuities if possible. The provided data on rates of complications does seem to suggest that this study merits larger follow-up studies to confirm your findings. You should add the specific software and packages you used to perform your statistics.

Answer: We thank the reviewer for the encouragement of our work. The reviewer is very correct and bang on the actual benefits of any surgical procedure for glaucoma being the preservation of long-term visual function. This is exactly the reason why we are now following the patients with visual fields in a separate longitudinal study comparing MIT with trabeculectomy and GATT. The visual fields are usually done on an annual basis in India which is why we did not feel adding a 6months visual field data, would be informative with regards to the long-term visual field progression rates with each surgical procedure. We definitely will update the reviewers and readers on the long-term results in our future papers or even on a personal request. We apologize for not mentioning the software used for analysis in this study. We have now added this information, as suggested. 

Reviewer #2: 1) Generally, I believe that the manuscript was well-written, clear, organized, and concise. I do not have much experience in statistical analysis so I am unable to comment on this matter, but from what I see, the tables, figures, and descriptions presented in the manuscript were clear and easily understandable. However, I think it may be more clear if the figures were located in the results section for easy reference (unless they are being considered as supplemental tables). If not placed in the results section, it would be easier to understand if legends were placed directly below the figures and tables.

Answers: We are very happy with the reviewer’s comments and pleased to share all the results of this new technique and any results of our future studies on this procedure on even a personal request. We would like to clarify that have made the figures separately available and not in the word document since adding them to the document decreases their resolution which is very crucial for understanding this procedure correctly. Also, the format of the manuscript is as per Plos One requirements which require files to be submitted separately with legends as per Plos One author instructions. 

2) There were a few grammatical/punctuational issues present, including:

Abstract:

"1day-1month" -> insert spaces to "1 day-1 month"

Materials & Methods:

"LV prasad eye institute" -> capitalize to "LV Prasad Eye Institute"

"followed up at 1 week, 4 weeks, 6months" -> insert space to "6 months"

Discussion:

"increasing wave of MIGS procedures in glaucoma management is understandable They" -> insert period to "is understandable. They"

"The Kahook dual blade (KDB) goniotomy procedure is has similar efficacy to other" -> reword to "goniotomy procedure has a similar efficacy to other"

Table 2:

"2mediciations" -> spelling error + insert space to "2 medications"

3) In both figures 1 and 3, the letters corresponding to specific to each image (A, B, C, etc.) could be larger and more clear to the reader. The letters seem to be a bit small and difficult to locate.

Overall, this is an exciting finding and authors did a great job with presentation of results and explaining details concisely. Some improvements to the value of this 

Answer: We thank the reviewer for the suggestions and all typos have been corrected in the revised manuscript. We have also made the letters in the figures bigger for easy visibility, as suggested.

---

## [Editor Report · Decision Letter 2]

7 May 2023

‘Microincisional trabeculectomy for glaucoma”

PONE-D-23-01259R2

Dear Dr. Rao,

We’re pleased to inform you that your manuscript has been judged scientifically suitable for publication and will be formally accepted for publication once it meets all outstanding technical requirements.

Kind regards,

Sanjoy Bhattacharya

Academic Editor

PLOS ONE
---

## [Editor Report · Acceptance letter]

12 May 2023

PONE-D-23-01259R2 

*‘Microincisional trabeculectomy for glaucoma”*  

Dear Dr. Rao:

I'm pleased to inform you that your manuscript has been deemed suitable for publication in PLOS ONE. Congratulations! Your manuscript is now with our production department. 

Kind regards, 

on behalf of

Dr. Sanjoy Bhattacharya 

Academic Editor

PLOS ONE